# Corrosion Behaviors of Artificial Chloride Patina for Studying Bronze Sculpture Corrosion in Marine Environments

Heehong Kwon 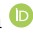

Department of Conservation and Art Bank, National Museum of Modern and Contemporary Art, Cheongju 28501, Republic of Korea; entasis@korea.kr

**Abstract:** Copper trihydroxychlorides, which are known as "bronze disease", are dangerous corrosion products that compromise the stability and conservation of bronze sculptures. Here, we performed artificial patina corrosion experiments on quaternary bronze (Cu-Zn-Sn-Pb) to examine the corrosion behavior of the chloride patina commonly found in bronze objects in marine environments. The chromaticity and reflectance of the patina in the context of the corrosion products indicate that copper trihydroxychloride, which is commonly found in a single color in marine environments, was produced early in the corrosion experiment. Furthermore, the corrosion of bronze had different effects on the alloying elements, contrary to pure copper corrosion. The chloride patina formed a single patina layer of copper trihydroxychlorides. This patina layer was divided into the outer porous powder and inner uniform layers. Furthermore, the interaction of oxygen in the atmosphere with the corrosion layer and internal oxidation of tin in the alloy promoted powdering. These results provide important basic data for research on sculpture conservation and corrosion characteristics, such as changes in color, chemical composition, and corrosion products on the patina surfaces of outdoor bronze sculptures.

**Keywords:** outdoor corrosion; patina; bronze disease; copper trihydroxychloride; atacamite; paratacamite; clinoatacamite; bronze sculpture





## 1. Introduction

The patina of outdoor bronze sculptures is considered a "passivation" layer, acting as a protective barrier from the atmospheric environment; however, other corrosion products are "activation" agents and are harmful to bronze sculptures [1]. The most well-known active corrosive agents are pitting corrosion and "bronze diseases", that is, copper trihydroxychlorides ($Cu_2Cl(OH_3)$), such as atacamite, paratacamite, and clinoatacamite [2]. Nantokite protects the inside of pitting by preventing diffusion of oxygen and the formation of the protective patina cuprite [3]. Moreover, owing to its low solubility, nantokite maintains the activity of the copper ions at a low level and promotes the anodic reaction of the metal [3]. Meanwhile, copper trihydroxychloride dissolves under acidic conditions, and the moisture layer of acidic groups on the surface of the patina (acid rain, acid snow) can cause instability [4]. Thus, it can cause internal physical stresses and weaken the bond owing to volume expansion and powdering [5].

Identification of potential "active corrosion" agents is one of the main roles of the conservator in the conservation treatment of bronze sculptures. Early identification of chloride patina allows us to take appropriate measures to prevent the progress of the corrosion [6]. The patina color of a bronze sculpture is important in terms of the permanence of the interaction of the work with the viewer and the intentions of the artist [7]. Hence, conservators should determine the original patina color and condition and consider how the patina has changed over time and to which extent this change will be acceptable to the public, collectors, artists, and foundations [6].

Particularly, copper trihydroxychlorides, such like atacamite, paratacamite, and clinoatacamite, have a highly porous and defective surface condition, which may cause dissolution because of condensation of acidic water between microcracks and crevices in the patina [8–12]. In addition, particulates such as dirt and dust in the atmosphere can be deposited on the porous surface, absorbing active ions and exacerbating corrosion [6,8]. Therefore, identifying the colors, chemical compositions, and corrosion properties of chloride patina surfaces, such as copper trihydroxychlorides, is important for conserving outdoor bronze sculptures.

A previous study examined the corrosion mechanisms of binary (Cu-Sn) and ternary (Cu-Sn-Pb) copper alloy systems. In addition, about 500 bronze sculptures made in Korea or other countries since the 20th century are made of quaternary alloys (Cu-Zn-Sn-Pb) [6,13–18]. Here, we conducted artificial patina corrosion experiments on quaternary bronze (Cu-Zn-Sn-Pb) with chemical composition and metallurgical attributes akin to those in 20th century outdoor bronze sculptures. The study aims to scrutinize the corrosion behavior of chloride patina, a prevalent occurrence in marine environments, as depicted in Figure 1. The changes in color, chemical composition, products corrosion, and patina growth on the surface of quaternary bronze (Cu-Zn-Sn-Pb) were investigated through these experiments.

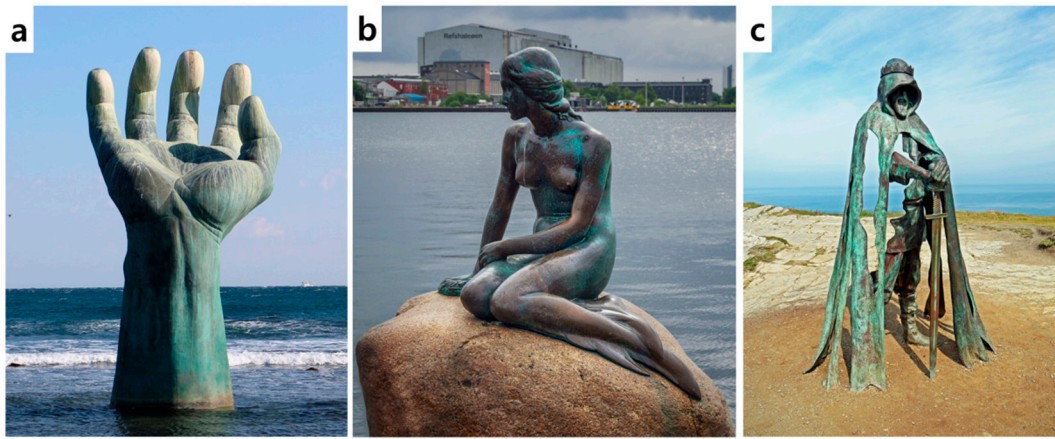

**Figure 1.** Outdoor bronze sculptures in marine environments: (**a**) Seungkook Kim's "Hand of Correlation (1999)" in Pohang, Korea [19]; (**b**) Edvard Eriksen's "The Little Mermaid (1913)" in Copenhagen, Denmark [20]; (**c**) Rubin Eynon's "Gallos (2016)" in Cornwall, UK [21].

## 2. Materials and Methods

The artificial patina corrosion experiments and analytical methods were identical to those reported in my previous study [22], except for the chloride patina corrosion solution.

### 2.1. Corrosion Experiment

Table 1 lists the composition of the bronze specimens created for the artificial patina study, which used the alloys found in an outdoor bronze statue and a previous experiment specimen. The specimens had dimensions of either 30 mm × 50 mm × 3 mm or 5 mm × 5 mm × 3 mm and were designed with perforations at the top for easy dipping into the corrosive solution during the experiments. The elimination of foreign substances was achieved by dipping the specimens in ethanol and cleaning them ultrasonically. Prior to this step, the samples were subjected to uniform abrasion using a #1000 sandpaper because the reflection of light depends on surface texture [23,24].

**Table 1.** Composition of the bronze specimen used for the corrosion experiment of the artificial patina.

| Composition (wt%) | | | | |
|---|---|---|---|---|
| **Cu** | **Zn** | **Sn** | **Pb** | **Total** |
| 88.8 | 5.1 | 3.1 | 3 | 100 |

An aqueous solution was created using 0.5 M copper(II) chloride ($CuCl_2$) and 0.1 M hydrochloric acid (HCl) in deionized water. This solution was prepared to induce corrosion products caused by chloride, a prevalent air pollutant in marine environments. The experiment involved immersing the specimen in the corrosive solution for 24 h, removing it, and allowing it to naturally dry for another 24 h under a relative humidity of 45%–55% at 20–24 °C. The process was repeated 50 times to gradually form a simulated chloride patina, as per previous studies [25–27]. Details of the experiment are outlined in Table 2.

**Table 2.** Corrosive solution used for producing the artificial chloride patina.

| Patina<br>Category | Chloride Patina |
|---|---|
| Corrosive solution | Aqueous solution of 0.5 M $CuCl_2$ and 0.1 M HCl (in deionized water) |
| Specimen fabrication | 50 cycles of (24 h deposition → 24 h natural drying) |

*2.2. Analyses*

### 2.2.1. Surface Condition and Form

To analyze the condition of the surface and shape of the artificial patina, the specimen was photographed using a digital camera (EOS 6D Mark II, Canon, Tokyo, Japan) and examined using a microscope (RH-2000, Hirox, Tokyo, Japan).

### 2.2.2. Chromaticity and Reflectance

To analyze the artificial patina color, the chromaticity and reflectance were numerically measured. Measurements were performed using a spectrum colorimeter (CR-400, Minolta, Tokyo, Japan) in the specular component excluded (SCE) mode with a measurement diameter of 3 mm. The surface reflectance was calculated as the average of five measurements in the visible light range (350–750 nm).

### 2.2.3. Products by Corrosion

To accurately analyze the crystal structure of the corrosion products on the artificial patina surface, X-ray diffraction (XRD) analysis (D8 Advance with DAVINCI, Bruker, MA, USA) was performed. A copper (Cu Kα 11.5418 Å) target was used with a current and accelerating voltage of 40 mA and 40 kV, respectively, in the X-ray tube. The analysis was performed with diffraction angles (2θ) in the range of 10°–90°, step size of 0.02°, and scan speed of 0.5 s/step.

Raman spectroscopy (LabRam ARAMIS, Horiba Jobin-Yvon, Kyoto, Japan) equipped with a 514 nm Nd:YAG laser and a 600 grooves/mm grating at a resolution of ~3.0 cm$^{-1}$, was applied to analyze the corrosion products present within each layer of the artificial patina. The analysis apparatus was configured to scan a range of 100–4200 cm$^{-1}$, with data analysis focused specifically on the 100–4000 cm$^{-1}$ interval, capturing pertinent corrosion product attributes.

### 2.2.4. Microstructure and Composition

The specimen was mounted on epoxy resin to examine the surface and cross-section of the artificial patina layer using scanning electron microscopy with energy-dispersive spectroscopy (SEM-EDS; JSM-6610LV, JP/X-Max, JEOL, Oxford, UK) in the secondary electron image mode at 15 kV with a spot size of 55 and working distance of 10 mm. A sequence of abrasion steps using sandpaper (#220–#4000) was applied to the specimen, followed by meticulous removal of surface scratches through fine abrasion (3 μm, 1 μm). Subsequently, the microstructure of the specimen was examined using a microscope (KH-7700, Hirox, Tokyo, Japan). The observed maximum and minimum measurements were indicative of the cross-sectional thickness of the patina layer.

## 3. Results

### 3.1. Microstructure and Material Characteristics of Specimen

Figure 2 shows the microstructure and uniform abrasion on the surface of the specimen before the corrosion experiment. Dendritic α phases were identified in the microstructure. This phase is often seen in sculptures made from Cu-Zn-Sn-Pb alloys [28–33] that have low tin concentrations. The observed microstructure resembled that of a cast structure which had not undergone any artificial processing, such as heat treatment [32].

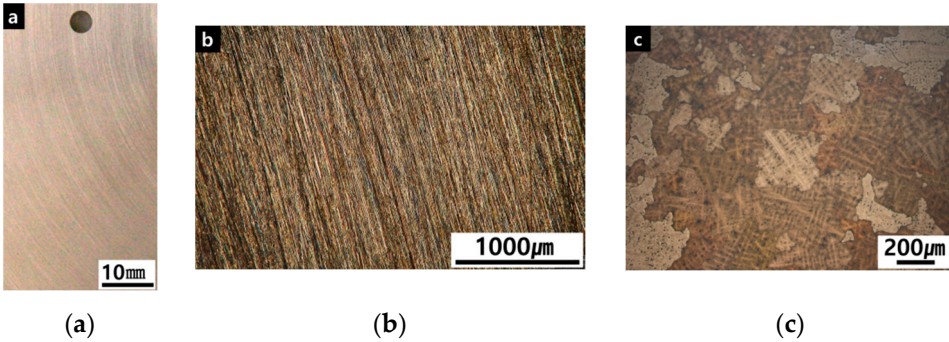

(a)                                     (b)                                     (c)

**Figure 2.** Surface and microstructure of the specimen before the artificial patina corrosion experiment: (**a**) Surface image of specimen. (**b**) Uniform abrasions of specimen surface. (**c**) Dendritic α phases were identified in the microstructure.

### 3.2. Surface Condition and Form

Figure 3 shows photographs of a bronze specimen treated with $CuCl_2$ and HCl aqueous solutions for 2400 h (50 cycles) to reproduce the corrosion products formed in a marine environment. The bronze specimen turned pale light blue after 48 h of corrosion (Cl1), was covered with a light blue patina at 96 h (Cl2), and then formed a uniform light blue patina layer until the end of the experiment at 2400 h (Cl50). The chloride patina produced in the corrosion experiment was porous and powdered, and many instances of delamination of the patina layer were observed during the deposition and drying of the corrosion solution. At the beginning of some of the corrosion experiments, a brown, powdered corrosion layer was generated; however, it was not observed after 288 h (Cl6). Previous studies indicated that a sulfide patina can take more than a year to form in outdoor environments, whereas a chloride patina takes as little as one month to form [33,34]. This rapid formation of chloride patinas is thought to cause weak bonding with the underlying layer and the formation of a porous patina layer.

### 3.3. Chromaticity and Reflectance

The chromaticity analysis values of the artificial patina surface were plotted in the color space of CIE L*a*b* (Figure 4a). The L* values ranged from 74.86 to 83.5, with a maximum deviation of 8.64. From 90 h (Cl2) to 2400 h (Cl50), that is, up to the end of the corrosion experiment, the change in brightness (L*) was small; the maximum deviation was 4.92. The a* value, which represents the red–green color, ranged from −18.5 to −2.96 and had a maximum deviation of 15.54. Except for Cl1 (−2.96), Cl2 to Cl50 were uniformly concentrated in the range of −18.5 to −14.45, which is consistent with the surface topography analysis results. The b* value, which represents the yellow–blue color, ranged from 3.61 to 8.57 and had a maximum deviation of 4.96. As the corrosion progressed, the b* value gradually decreased; that is, the color became closer to blue.

The reflectance measurements of the chloride artificial patina surface (Figure 4b) yielded the same spectral shape, except Cl1 (48 h). The pale light blue patina Cl1 (48 h) exhibited lower reflectance values in the blue wavelength region (380–500 nm) and higher reflectance values in the green (500–565 nm) and yellow (565–625 nm) wavelength regions. Contrarily, the light blue patina from Cl2 (90 h) to Cl50 (2400 h) exhibited a low reflectance in the blue region (380–500 nm) and a sharp increase in reflectance in the green region

(500–565 nm). Moreover, it exhibited lower reflectance values in the yellow (565–625 nm) and red (625–750 nm) regions. This was consistent with the outcomes of chromaticity analysis, which confirmed that the chloride patina formed rapidly in a short time and did not change in color.

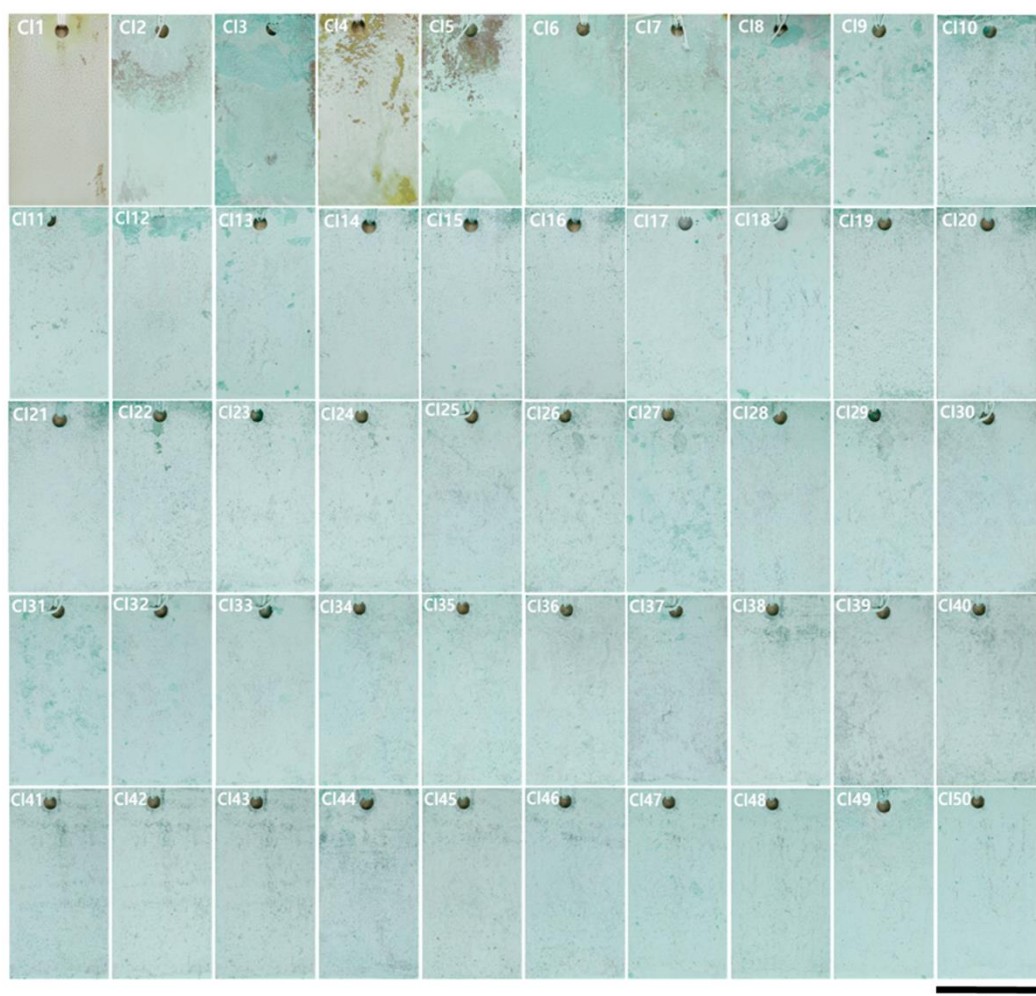

**Figure 3.** Corrosion experiment involving the artificial chloride patina: photographs of a bronze specimen treated with $CuCl_2$ and HCl aqueous solutions for 2400 h (50 cycles) to reproduce the corrosion products formed in a marine environment.

### 3.4. Analysis of Products by Corrosion

Corrosion product identification and semiquantitative analysis of the artificial patina surface were performed using XRD. Semiquantitative analyses were performed eight times in total from Cl1 (at the onset of the color change) to Cl50 (at the end of the experiment) to quantitatively analyze the changes in the contents of corrosion products (Figure 5). The XRD analysis revealed three major corrosion products: nantokite, atacamite, and paratacamite. Moreover, zincite ($Cu_3Zn$), zinc chlorate ($Zn(ClO_3)_2$), and ammonium chlorate ($NH_4ClO_3$) were identified.

When the evolution of the corrosion products was examined, 84.1% nantokite and 13.7% atacamite were detected in Cl1 (48 h), which changed to a pale light blue color at the beginning of the corrosion experiment. At Cl5 (240 h), the nantokite content was 70.4%, indicating a 13.7% decrease from Cl1, and the atacamite content was 32.5%, indicating a 56.2% increase. Additionally, paratacamite was first detected, with a content of 23.8%.

Nantokite was not identified after Cl10 (480 h), indicating that nantokite, a precursor of atacamite, was replaced by atacamite. At Cl10 (480 h), atacamite and paratacamite were

identified, with contents of 56.4% and 30.8%, respectively, and from C20 (960 h) onward, the paratacamite and atacamite contents remained between 70.0% and 80.6% and between 13.9% and 19.8%, respectively. At Cl50 (2400 h), patina surfaces with paratacamite and atacamite were formed.

According to the Raman spectrum of the chloride specimen surface (Figure 6), Cl1 (48 h), which formed a light blue patina layer, was identified as clinoatcamite ($Cu_2Cl(OH)_3$), one of the copper trihydroxychlorides, because the Raman shifts at 122, 153, 363, 506, 809, 979, 3351, and 3434 $cm^{-1}$ at the green and white spots were almost identical, and the peaks had similar shapes. From Cl5 (240 h) to Cl50 (2400 h), when a light blue patina layer was formed, green, dark green, and white regions were identified as clinoatacamite because the Raman shifts at 122, 153, 363, 506, 809, 979, 3351, and 3434 $cm^{-1}$ were almost identical, and the peaks had similar shapes.

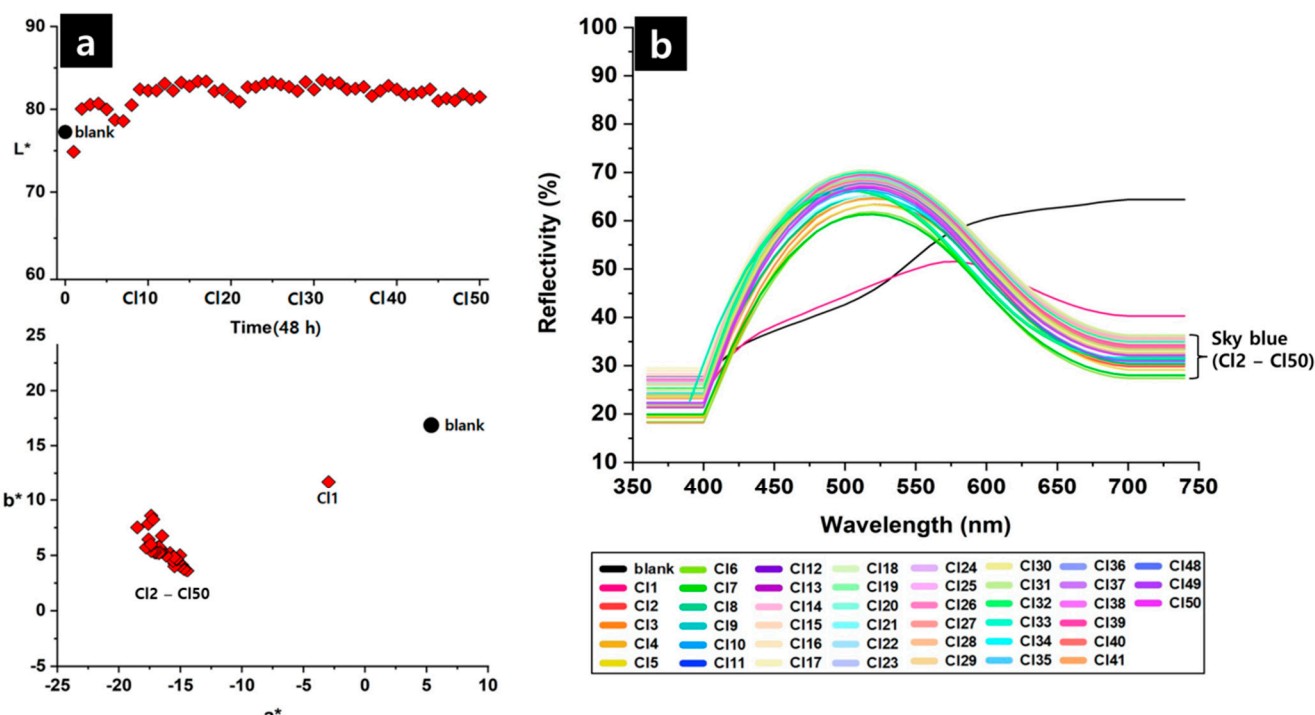

**Figure 4.** (**a**) Chromaticity and (**b**) reflectance analysis results for the artificial chloride patina: (**a**) The L* values ranged from 74.86 to 83.5, the a* values ranged from −18.5 to −2.96, and the b* values ranged from 3.61 to 8.57. (**b**) The reflectance measurements of surface yielded the same spectral shape, except for Cl1 (48 h).

Clinoatacamite was identified in the Raman analysis, whereas atacamite and paratacamite were identified in the XRD analysis. Because of the similar face spacings (d-values) in the crystallographic data of paratacamite and clinoatacamite, it was difficult to accurately identify the two corrosion products using XRD [31]. Furthermore, because many outdoor bronze sculptures contain large amounts of zinc, some of the reports on paratacamite may be accurate, and they may coexist with clinoatacamite as a corrosion product in mineral form [2].

### 3.5. Microstructures and Compositions of Corrosion Products

SEM-EDS analysis confirmed the chemical composition and microstructure of the corrosion products by layer, and the changes in alloy composition owing to corrosion behavior were examined using EDS (Figure 7, Table 3). In addition, the corrosion products of each layer were examined through Raman analysis. Magnified surface and cross-sectional views of the chloride artificial patina and the growth stages are shown in Figure 8.

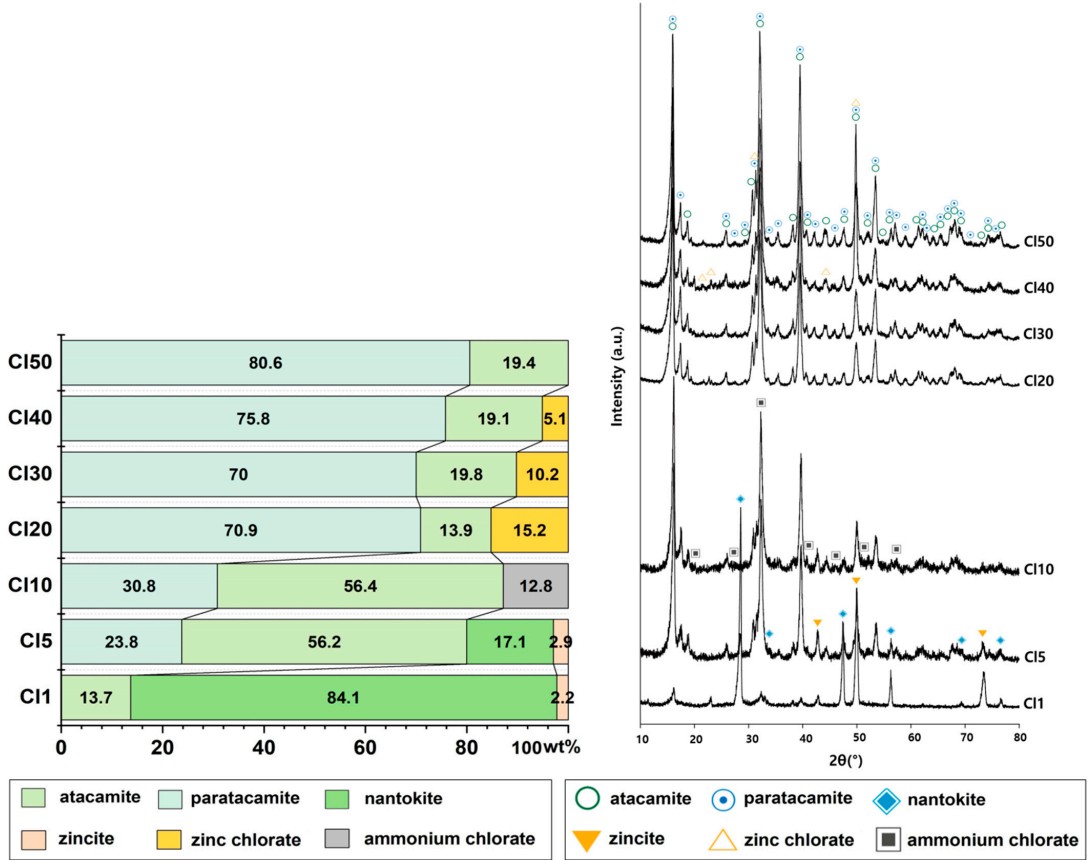

**Figure 5.** Semiquantitative XRD analysis results for the artificial chloride patina surface. The XRD analysis revealed three major corrosion products: nantokite, atacamite, and paratacamite. Moreover, zincite ($Cu_3Zn$), zinc chlorate ($Zn(ClO_3)_2$), and ammonium chlorate ($NH_4ClO_3$) were identified.

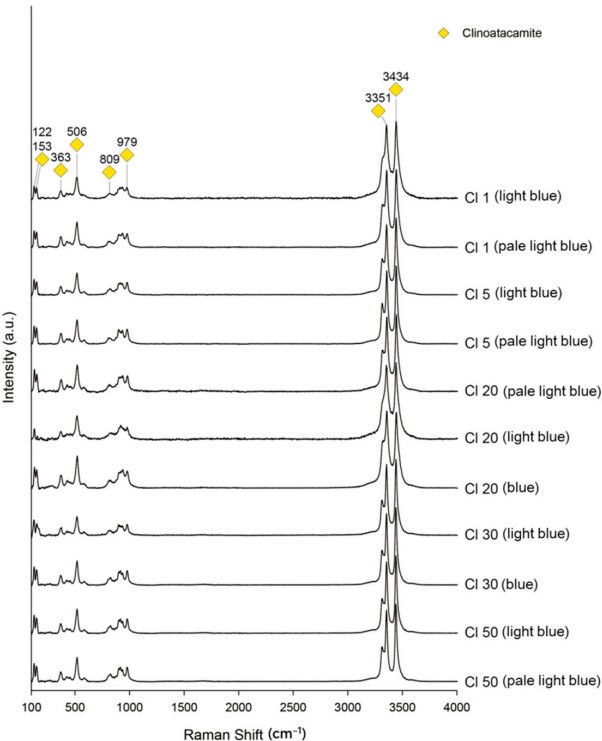

**Figure 6.** Raman analysis results for the artificial chloride patina surface.

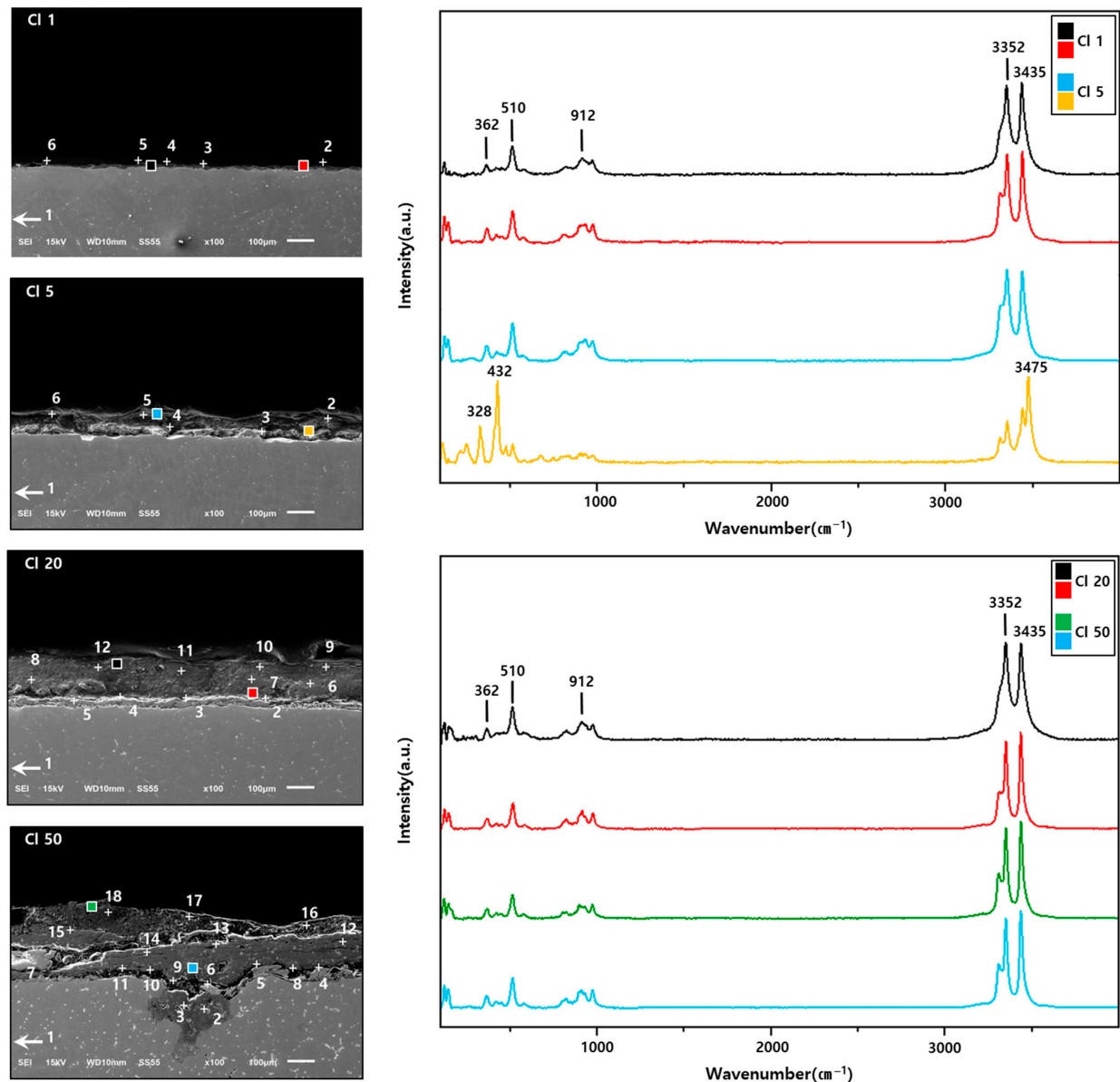

**Figure 7.** SEM-EDS and Raman results indicate the locations of the chloride artificial patina: Cl1, Cl5, Cl20, and Cl50.

The cross-sectional observation of Cl1 (48 h) to Cl50 (2400 h) revealed that the thickness of the patina increased by 2–10 μm (Cl1), 70–130 μm (Cl5), 110–140 μm (Cl20), and 200–310 μm (Cl50). As indicated in Figure 8, porous crystals were formed on the surface in a disordered manner and flaking of the surface layer occurred. This is attributed to the porous properties of the patina and the inclusion of nantokite within its patina [35,36]. In the cross-section, an outer powder and inner uniform layers were identified.

The EDS analysis indicated high contents of copper (18%–56%), oxygen (15%–72%), and chlorine (approximately 5%–31%) in the corrosion layer. Furthermore, the XRD and Raman analysis of the artificial patina surface confirmed that it was composed of copper hydroxychlorides, that is, atacamite, paratacamite, and clinoatacamite. There was a significant difference in tin content between the outer porous powder (Cl1–Cl50: average of 0.46%) and inner homogeneous layers (Cl20–Cl50: average of 14.9%). The EDS analysis results indicated that the contents of copper (85.61→84.87 wt%) and zinc (7.93→7.79 wt%) decreased during the corrosion process, whereas the contents of Sn (3.57→3.78 wt%) and lead (1.38→2.87 wt%) increased.

**Table 3.** SEM-EDS results for the artificial chloride patina.

| No. | Position | Composition (wt%) | | | | | | Layer |
|-----|----------|------|------|------|------|------|------|-------|
| | | **Cu** | **Zn** | **Sn** | **Pb** | **O** | **Cl** | |
| Cl1 | 1 | 85.61 | 7.93 | 3.57 | 1.38 | 1.49 | 0.02 | bronze |
| | 2 | 42.85 | 0.55 | - | 9.24 | 26.63 | 20.73 | |
| | 3 | 41.77 | 1.76 | 0.17 | 6.57 | 29.01 | 20.72 | |
| | 4 | 38.34 | 1.34 | 0.01 | 7.00 | 33.24 | 20.06 | outer powder layer |
| | 5 | 29.85 | 1.52 | - | 6.03 | 47.02 | 15.58 | |
| | 6 | 42.70 | 0.71 | 0.15 | 1.71 | 30.49 | 24.23 | |
| Cl5 | 1 | 85.51 | 8.07 | 3.47 | 1.53 | 1.34 | 0.09 | bronze |
| | 2 | 40.56 | 0.08 | 1.30 | 1.03 | 51.91 | 5.12 | |
| | 3 | 21.52 | - | - | - | 72.76 | 5.72 | |
| | 4 | 31.05 | 0.46 | 0.06 | 2.88 | 37.2 | 28.35 | outer powder layer |
| | 5 | 18.70 | 0.57 | - | 0.84 | 70.92 | 8.98 | |
| | 6 | 42.17 | - | - | 4.83 | 40.63 | 12.37 | |
| Cl20 | 1 | 85.29 | 7.95 | 4.07 | 1.55 | 1.09 | 0.04 | bronze |
| | 2 | 24.57 | 0.15 | 23.59 | 6.02 | 15.81 | 29.86 | |
| | 3 | 47.32 | 0.84 | 9.77 | 0.79 | 24.51 | 16.77 | |
| | 4 | 33.63 | 0.84 | 25.46 | 0.76 | 24.76 | 14.54 | |
| | 5 | 40.56 | 1.91 | 22.99 | 4.16 | 20.31 | 10.08 | internal uniform layer |
| | 6 | 46.84 | 0.53 | 9.25 | 2.12 | 20.35 | 20.92 | |
| | 7 | 32.86 | 1.15 | 18.98 | 1.08 | 26.71 | 19.23 | |
| | 8 | 40.85 | 0.68 | 12.12 | 1.15 | 19.85 | 25.36 | |
| | 9 | 49.9 | - | 0.02 | 2.88 | 25.94 | 21.26 | |
| | 10 | 37.04 | - | 0.82 | 6.63 | 24.47 | 31.04 | |
| | 11 | 42.76 | - | 0.64 | 3.43 | 31.14 | 22.03 | outer powder layer |
| | 12 | 29.54 | - | 0.09 | 1.45 | 16.60 | 18.75 | |
| Cl50 | 1 | 84.87 | 7.79 | 3.78 | 2.87 | 0.56 | 0.13 | bronze |
| | 2 | 50.23 | - | 6.64 | 1.3 | 21.04 | 20.8 | internal uniform layer |
| | 3 | 56.72 | - | 0.32 | - | 22.82 | 20.14 | |
| | 4 | 89.58 | 8.51 | 1.43 | - | 0.39 | 0.09 | |
| | 5 | 85.93 | 7.59 | 6.08 | - | 0.31 | 0.09 | alloy molten layer |
| | 6 | 85.82 | 7.44 | 2.81 | 0.25 | 3.05 | 0.63 | |
| | 7 | 83.11 | 7.44 | 3.35 | 1.13 | 4.27 | 0.69 | |
| | 8 | 57.94 | 3.92 | 4.69 | 5.46 | 17.81 | 10.18 | |
| | 9 | 52.54 | 2.03 | 1.50 | 1.64 | 20.30 | 22.00 | boundary between microstructure and corrosion layer |
| | 10 | 53.73 | 3.65 | 2.21 | 1.25 | 24.24 | 14.91 | |
| | 11 | 57.70 | 3.17 | 4.65 | 3.11 | 17.57 | 13.80 | |
| | 12 | 42.65 | - | 17.52 | 1.89 | 21.39 | 16.55 | |
| | 13 | 42.34 | 0.02 | 16.05 | 1.54 | 18.93 | 21.13 | |
| | 14 | 32.18 | 0.81 | 21.77 | 4.29 | 21.29 | 19.66 | internal uniform layer |
| | 15 | 46.64 | - | 9.35 | 1.81 | 19.90 | 22.31 | |
| | 16 | 54.99 | - | - | 0.53 | 20.87 | 23.61 | |
| | 17 | 51.94 | - | 0.51 | 0.34 | 22.13 | 25.08 | outer powder layer |
| | 18 | 50.41 | 0.33 | 1.31 | 1.44 | 20.56 | 25.97 | |

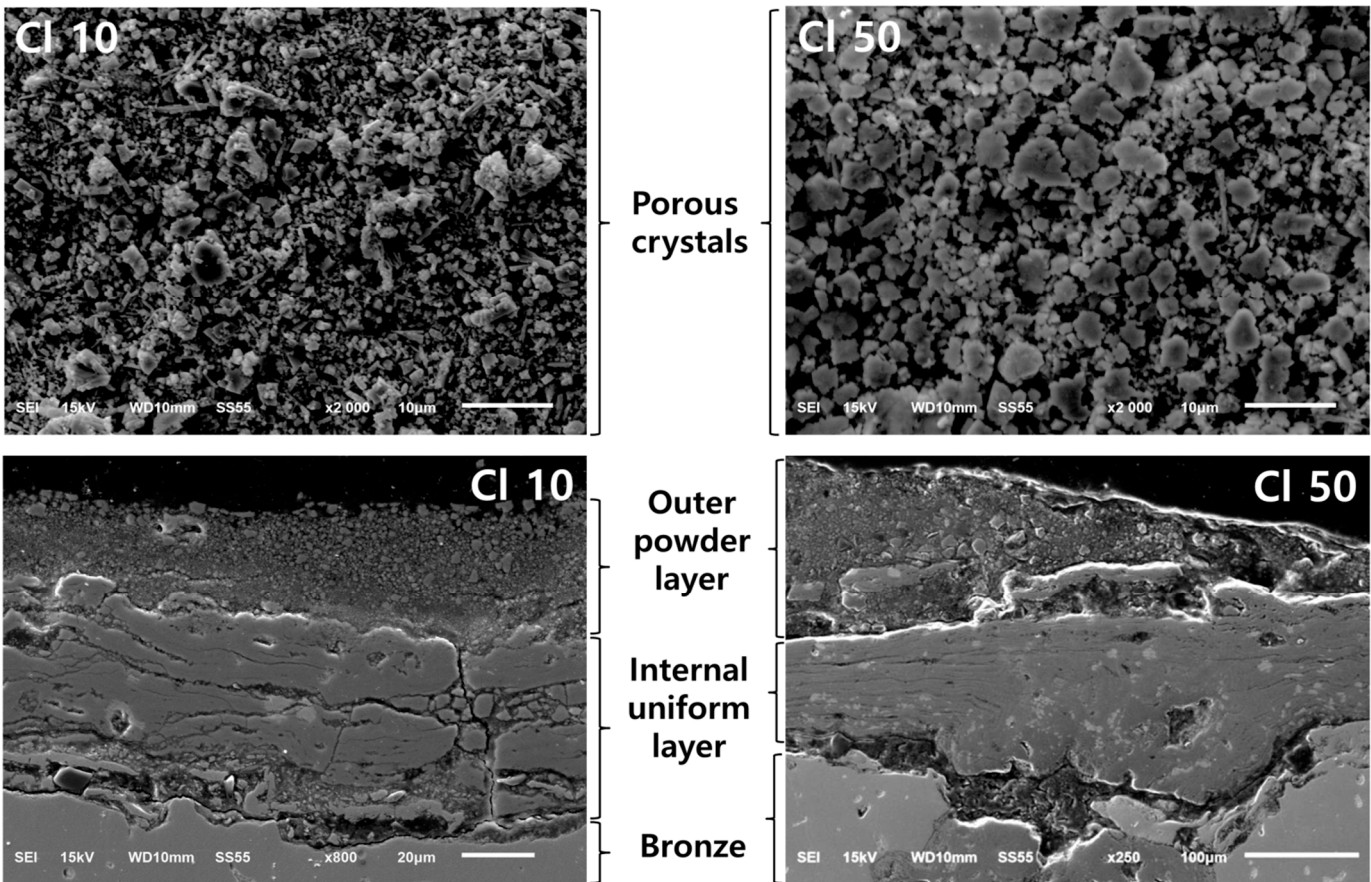

**Figure 8.** SEM images of the surface and cross-section of the chloride artificial patina. Porous properties were observed on the surface of the patina. In the cross-section, the outer powder and inner uniform layer were identified.

As a result of Raman analysis of the cross-sections of the C1, Cl5, Cl20, and Cl50 specimens, it was confirmed that clinoatacamite was present as the result of Raman analysis on the surface. In addition, the corrosion products of the outer and inner layers were the same.

## 4. Discussion

The corrosion characteristics and chemical composition changes of outdoor bronze sculptures were investigated by corrosion experiments of a chloride artificial patina in a marine environment. Moreover, a detailed analysis of the formation process of this chloride patina was undertaken. Particularly, an artificial patina corrosion experiment was conducted to obtain standard data on corrosion products such as copper trihydroxychlorides, which are identified in the corrosion characteristics of outdoor bronze sculptures. However, according to several previous studies [4,37–39], products of multiple corrosion are commonly detected in outdoor sites, sometimes in combination with surrounding soil minerals and contaminants, such as soot and ash.

### 4.1. Color Classification and Corrosion Products of Chloride Patina

According to the correlations between the surface color change of the chloride artificial patina during the corrosion process and the measurable change in the corrosion products quantities, the patinas were classified into one color based on chromaticity ($\triangle E^*$) and reflectance (Figure 9). In the initial stages of the corrosion experiment (Cl1–Cl2), the color-difference value was approximately 16 (0→16), and it increased significantly over a short time (96 h). However, it remained constant for a long period from Cl5 to Cl50 (2344 h), with a color difference of $\leq 4$ (0.3→4.9). These results can be observed more clearly in the color

difference a*, which represents the blue–green color. The color difference a* for Cl1–Cl2 was approximately 15, and it increased significantly over a short period; however, it remained at approximately 4 over the long period from Cl2 to Cl50.

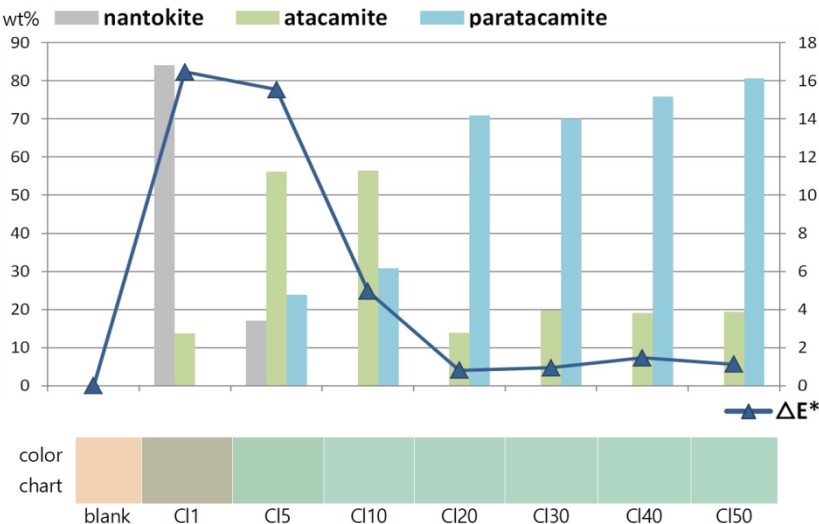

**Figure 9.** Chromaticity comparison graph of corrosion products of the artificial chloride patina.

The reflectance was characterized by inflection points in the spectrum at approximately 420 and 580 nm early in the corrosion experiment (Figure 4b). As the corrosion progressed from Cl1 to Cl2, there was a sharp increase in reflectance in the blue region of the 420 nm spectrum and a large increase of 10%–30% in the green (500–565 nm) region. The reflectance decreased sharply in the yellow region of the 580 nm spectrum, and it exhibited a large decrease of approximately 10% in the red (625–750 nm) region.

Cl1 is the point at which the pale light blue patina began to appear early in the corrosion experiment in visual and microscopic observations. Through XRD analysis, the initial atacamite, that is, copper chloride, was identified (13.7%). Thus, the single-color classification of chloride artificial patina is consistent with the characteristics of patina typically produced in marine atmospheric environments.

The color-difference values of the chloride artificial patina were compared with the measurable change in the corrosion products quantities. The nantokite content decreased sharply at Cl5 (240 h) after initial formation, and nantokite was not detected thereafter, with a proportional decrease in the color-difference value. The changes in the contents of atacamite (light blue) and paratacamite were >80% after Cl5, and the color-difference value remained almost constant over time, indicating an inverse correlation.

### 4.2. Composition and Corrosion Products of Chloride Patina

According to the correlation between the corrosion products and chemical composition of the chloride artificial patina during corrosion (Figure 10), pale light blue nantokite (approximately 84%) and atacamite (approximately 13%) were the main corrosion products at the beginning of the corrosion experiment (Cl1: 48 h). However, as the corrosion progressed, the nantokite content decreased, as nantokite was replaced by copper trihydroxychloride ($Cu_2Cl(OH_3)$) at Cl10 (480 h). Atacamite was detected throughout the corrosion process. Meanwhile, paratacamite was first observed at Cl5 (240 h), and the paratacamite content increased to 80.6% at the end of the experiment (Cl50: 2400 h). Thus, a chloride patina is a highly porous crystalline structure that has a weak bond with the underlying layer owing to its rapid generation.

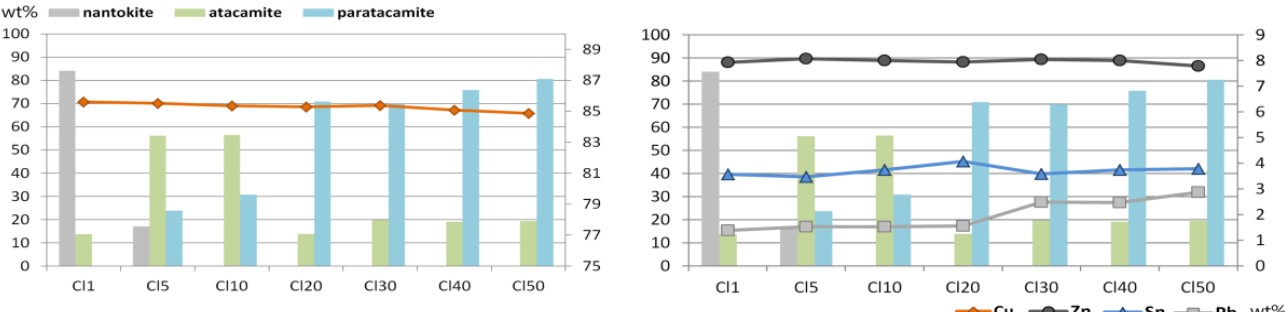

**Figure 10.** Correlation between the composition and corrosion products of the artificial chloride patina during the corrosion process: The contents of copper and zinc decreased in both the patina and inner alloy, suggesting that the formation of the patina is based on the selective dissolution/leaching of these two elements. The enrichment of tin and lead is explained by the low solubility and high stability of tin oxide.

The SEM-EDS surface analysis revealed that during the corrosion process, the contents of tin and lead increased, while the contents of copper and zinc decreased (Figure 10). This confirmed two types of corrosion behaviors concerning the distribution of alloy elements in the patina. The contents of copper and zinc decreased in both the patina and inner alloy, suggesting that the formation of the patina is based on the selective dissolution/leaching of these two elements. The enrichment of tin and lead is explained by the low solubility and high stability of tin oxide.

In tetragonal bronzes (Cu-Zn-Sn-Pb), lead exhibits non-uniform corrosion, oxidizing before the other elements and retarding the oxidation of zinc and copper [40]. This is caused by the galvanic coupling between the different alloys. Because lead is not mixed in the solid solution (Cu-Zn-Sn), the lead globules within the alloy function as sacrificial anodes that oxidize first.

### 4.3. Corrosion Mechanism of Chloride Patina

The formation process and atmospheric corrosion mechanism of the chloride patina were investigated through artificial patina corrosion experiments, as shown in Figure 11. The chloride patina formed a monolithic patina layer of copper trihydroxychlorides ($Cu_2Cl(OH)_3$), such as atacamite, paratacamite, and clinoatacamite. Although nantokite was identified by XRD analysis at the beginning of the corrosion experiment, SEM observations did not reveal differences between the layers.

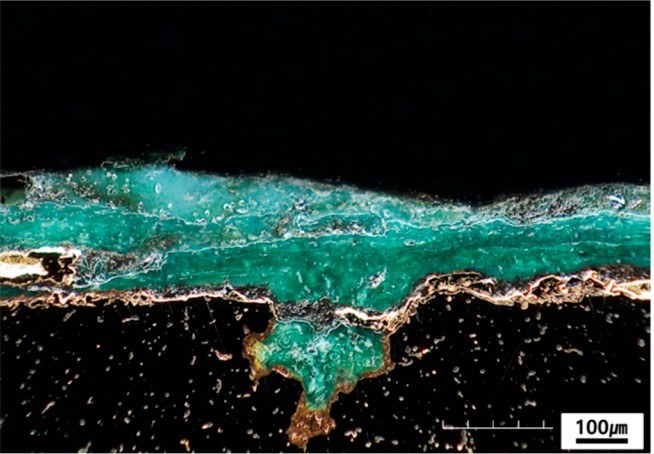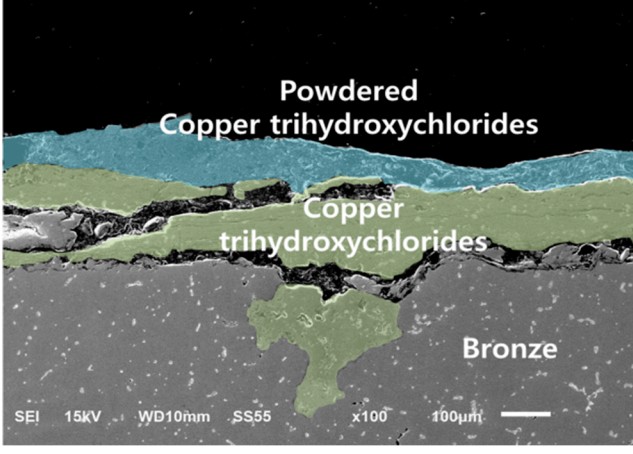

**Figure 11.** Chloride patina (Cl50): (**left**) microscopic (dark-field) image; (**right**) schematic of the double layers based on SEM images. The chloride patina formed a single patina layer of copper trihydroxychlorides. This patina layer was divided into outer porous powder and inner uniform layers.

Fitzgerald et al. [37,41] reported that the oxidation process of cuprite should be slower than that of copper, and if the corrosion of the metals were faster, the cuprite layer would disappear. Furthermore, in a corrosion experiment involving excavated bronze artifacts, as the concentration of Cl⁻ ions increases, the hydrolysis reaction of nantokite accelerates, which shortens the formation stage of cuprite, and when the Cl⁻ concentration increases further, the hydrolysis of nantokite omits cuprite and forms copper trihydroxychlorides, such as atacamite [42]. The chloride artificial patina corrosion experiment of this study suggests that the yellowish metallic luster between the corrosion layer and base metal (micrograph shown on the left side of Figure 11) was caused by the dissolution of the interface of the base metal owing to the high concentration of $CuCl_2$ and HCl aqueous solutions, thus omitting the formation of cuprite.

Figure 12 illustrates the formation process of chloride patina and the atmospheric corrosion mechanism. The corrosion process of the chloride patina forms nantokite (CuCl) through the interaction of $Cu^+$ and $Cl^-$ ions from dissolved cuprite ($Cu_2O$) in an environment with a high RH and oxygen concentration. Nantokite is stable only in the absence of oxygen and moisture under acidic conditions. The unstable nantokite crystals function as a precursor for atacamite formation through many subsequent dissolution-ion pairing-precipitation steps [26,35]. The formation of nantokite in the presence of oxygen and humidity results in a cyclic corrosion process, producing a light blue powder layer of copper trihydroxychloride. The Gibbs free energies of the four polymorphs produced (i.e., botallackite, paratacamite, atacamite, and clinoatacamite) were similar: −1322.6, −1338.8, −1339.2, and −1341.8 kJ/mol, respectively. Thermodynamic data suggest that the most stable polymorph is clinoatacamite. Clinoatacamite is the final product, while the other phase is intermediate [35,43,44].

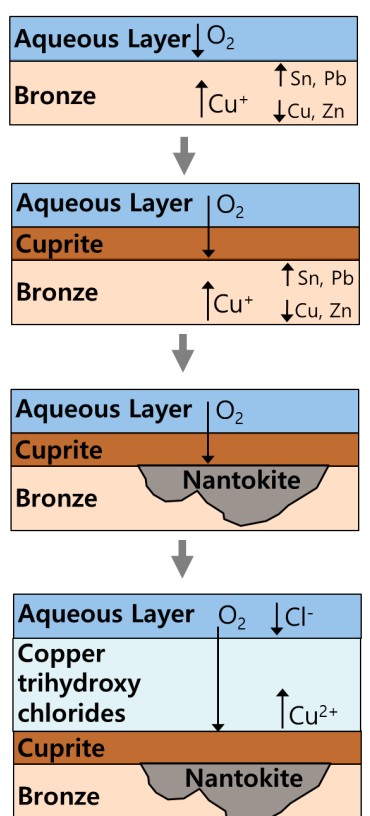

•**Cu, Zn**: Selective dissolution according to decuprification and dezincification

•**Sn**: Formation of patina layer from condensed insoluble corrosion products

•**Cuprite, Nantokite**: The chloride patina forms nantokite (CuCl) through the interaction of $Cu^+$ and $Cl^-$ ions from dissolved cuprite ($Cu_2O$) in an environment with a high RH and oxygen concentration.

•**Copper trihydroxychloride**: The formation of nantokite in the presence of oxygen and humidity results in a cyclic corrosion process, producing a light blue powder layer of copper trihydroxychloride.

**Figure 12.** Schematic of the corrosion mechanism of chloride patina.

Corrosion behavior is accompanied by a volume expansion of the corrosion products within the corrosion layer, which is commonly known as bronze disease. Moreover, paratacamite has a larger molal volume (61.02 cm$^3$/mol) than nantokite (23.88 cm$^3$/mol).

Consequently, the conversion of nantokite into copper trihydroxychlorides, such as parata-camite, is correlated with significant volume expansion. In the corrosion experiment of this study, the thickness of the patina layer gradually increased (Figure 13; Cl1: 2–10 μm, Cl5: 70–130 μm, Cl10: 70–100 μm, Cl20: 110–140 μm, Cl30: 110–200 μm, Cl40: 200–220 μm, Cl50: 200–310 μm). This was attributed to the volume expansion of copper trihydroxychlorides. Therefore, the continuous volume expansion of the chloride patina deforms the external appearance of the bronze sculpture and causes internal physical stresses, which can lead to cracks by weakening the bonding force between the inner and outer layers [5,45].

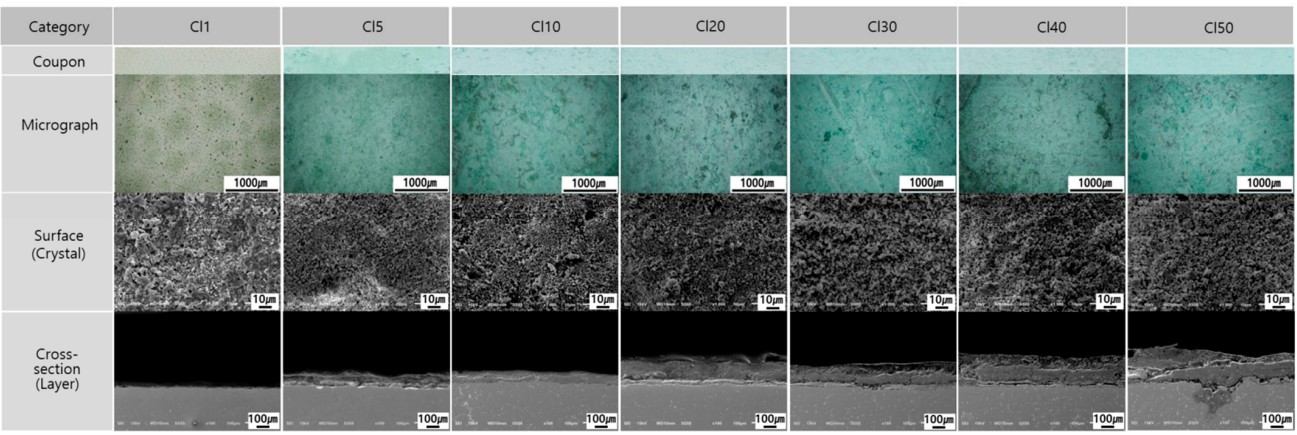

**Figure 13.** Growth stages of the chloride patina: Regarding the microstructure of the chloride patina, a single patina layer of copper trihydroxychlorides was formed, and an orderly volume expansion was observed. This can cause internal physical stress and cracking by weakening the bonding force between the inner and outer layers.

In the chloride patina corrosion experiments, the copper trihydroxychloride was divided into outer powder and inner uniform layers, as shown in Figure 14, respectively. The outer powder layer was characterized by a high concentration of oxygen at the outermost boundary with the atmosphere, as confirmed by EDS mapping. The interaction of moisture and oxygen promotes porosity and powdering [4,46], and elemental oxygen was detected in this study. Furthermore, the tin content was high in the inner homogeneous layer (Cl20–Cl50: average of 14.9%), and low in the outer porous powder layer (Cl1–Cl50: average of 0.46%). This was caused by a corrosion mechanism identified by decuprification or leaching of copper. The internal oxidation of tin in the alloy was identified as one of the factors promoting powdering [43].

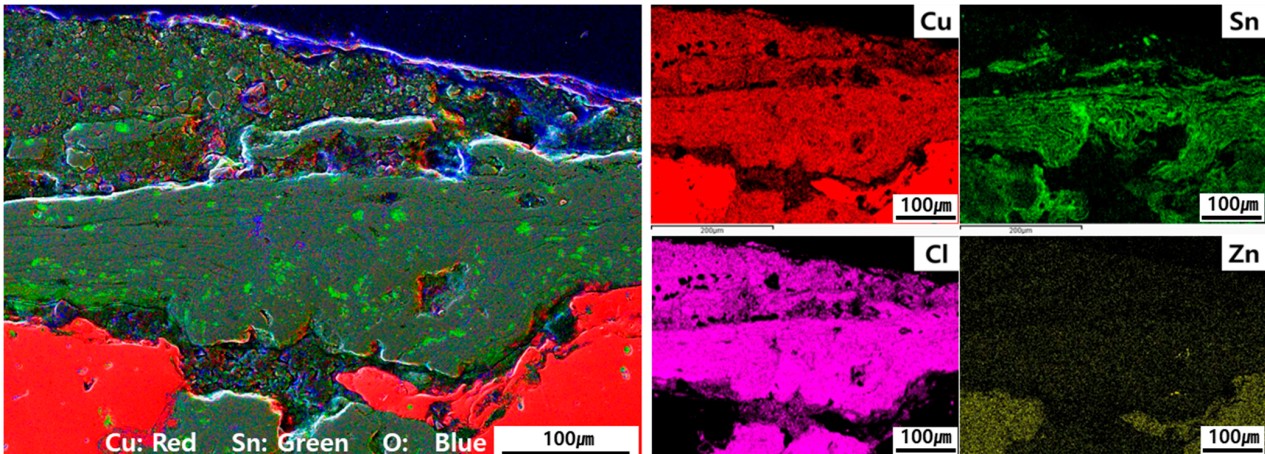

**Figure 14.** SEM mapping of the chloride artificial patina (Cl50): The interaction of atmospheric oxygen with the corrosion layer and the internal oxidation of tin in the alloy promoted powdering.

A comparison and analysis of the corrosion characteristics of the sulfide and chloride patinas commonly found on outdoor bronze sculptures revealed that sulfide patinas are characterized by the migration of copper ions to the outermost layer. This results in a denser patina layer and the continuous growth of green brochantite crystals [22]. Conversely, a chloride patina is a highly porous crystalline structure that has a weak bond with the underlying layer owing to its rapid generation. Furthermore, because it is accompanied by volume expansion, the patina should be detected and conserved early, and the corrosion products should be monitored. The sulfide patina, which imparts aesthetic and historical values, should be conserved as it forms a protective layer. However, the chloride patina is the main active corrosion factor that damages bronze sculptures and thus should be treated to eliminate or stabilize corrosion products. In the case of outdoor bronze sculptures, lacquer, wax, and corrosion inhibitors are used as coating agents. In particular, with regular maintenance and good application, wax can be an effective coating, as attested by years of study and use on outdoor sculpture collections around the world [47].

## 5. Conclusions

Corrosion experiments were performed on a chloride artificial patina to characterize the corrosion of outdoor bronze sculptures in marine environments. The chromaticity and reflectance of the patina and corrosion products show that the chloride patina was largely of one color, particularly atacamite, which was one of the copper trihydroxychlorides observed early in the corrosion experiment. The single color of the chloride artificial patina is consistent with the typical characteristics of patinas produced in marine environments.

The atmospheric corrosion of the quaternary bronze in our corrosion experiments reduced the contents of the base metals copper and zinc; however, tin and lead contents increased. The lower copper and zinc contents were attributed to the decuprification and dezincification inside the bronze due to selective dissolution. The cations moved from the inside of the alloy through the patina toward the interface with the atmosphere, and tin is concentrated as an insoluble corrosion product, forming a patina layer and playing a stabilizing role. Although bronze corrosion in the atmosphere was thought to react similarly to pure copper corrosion, the corrosion process of bronze has been shown to have a different effect on alloying elements.

Regarding the microstructure of the chloride patina, a single patina layer of copper trihydroxychlorides, such as atacamite, paratacamite, and clinoatacamite, was formed, and an orderly volume expansion was observed. This can cause internal physical stress and cracking by weakening the bonding force between the inner and outer layers. Furthermore, the copper trihydroxychloride was divided into outer powder and inner uniform layers, and the interaction of atmospheric oxygen with the corrosion layer and the internal oxidation of tin in the alloy promoted powdering.

The characteristics of the chloride patina were investigated according to its corrosion behavior. However, through further studies, such as electrical impedance spectroscopy measurements and gravimetric analysis, quantitative data on chloride patina will be presented. The findings of this study can be used to investigate the conservation and corrosion characteristics of outdoor bronze sculptures.

**Funding:** This research was funded by the Conservation Science Research Project of the National Museum of Modern and Contemporary Art, Republic of Korea (MMCA).

**Institutional Review Board Statement:** Not applicable.

**Informed Consent Statement:** Not applicable.

**Data Availability Statement:** Not applicable.

**Conflicts of Interest:** The author declares no conflict of interest.

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
