# Peer review of "Corrosion Behaviors of Artificial Chloride Patina for Studying Bronze Sculpture Corrosion in Marine Environments"

_coatings, doi:10.3390/coatings13091630_

Round 1

Reviewer 1 Report

Coatings-2595774

1-      The authors need to focus more on quantitative information, not qualitative ones. Moreover, all abbreviations should be defined for the first time.

2-      In the "Introduction" section, the authors were supposed to clarify the answers to the following questions;
a. Which exact problem was supposed to be solved by the present research?
b. Which new achievement(s) was supposed to be obtained by the present research compared to the previous reports?

3-      Fig. 2 requires legend and each one should be explained. Further explanation regarding the OM image of the specimen before the artificial patina corrosion experiment should be presented.

4-      Fig. 3 requires scale bar and scale value. In addition, the legend and corresponding definition should be provided in the caption of the figure. Moreover, the quality of the figure should be improved.

5-      Fig. 4 and Fig. 5 as well as Fig. 12 require legend and the legend and corresponding definition should be provided in the caption of the figure. In addition the scale bar and scale value of the SEM image is not clear (Fig. 9).

6-      This study suffers from poor quantitative data presentation such as electrochemical impedance spectroscopy measurements and reports the value of charge transfer resistance, solution resistance, constant phase element, and double layer capacitance. Alternatively the author could provide a potentiodynamic polarization test and report the value of corrosion potential, current density, corrosion rate for all samples.

7-      The author presents a correlation between the composition and corrosion products of the artificial chloride patina during the corrosion process. Further description regarding the measurement of the quantitative measurement of corrosion products and its relation with the composition should be presented. I strongly suggest that the author measure the corrosion rate of all samples by conducting weight loss tests.

8-      The author presents the corrosion mechanism of chloride patina in section 4.4. in this regard, For the benefit of the readers the authors could present the schematic corrosion mechanism of the bronze-based material in an aggressive environment.

9-      A reference about corrosion behavior of Cu-based alloy may be useful for this article: Transactions of Nonferrous Metals Society of China 25 (2015) 1158-1170. In addition, surprisingly small references to the Coatings in the literature despite the large relevant literature there. This should be improved. There are several important papers in recent literature.     

Author Response

Thank you very much for reviewing the article.
I revised and supplemented the parts you reviewed as much as possible.

Review 1 Report (Round 1)

1- The authors need to focus more on quantitative information, not qualitative ones. Moreover, all abbreviations should be defined for the first time.

   Modification complete.

2- In the "Introduction" section, the authors were supposed to clarify the answers to the following questions;

  1. Which exact problem was supposed to be solved by the present research?
  2. Which new achievement(s) was supposed to be obtained by the present research compared to the previous reports?

The contents of the pointed part were further reinforced.

3- Fig. 2 requires legend and each one should be explained. Further explanation regarding the OM image of the specimen before the artificial patina corrosion experiment should be presented.

Modification complete.

4- Fig. 3 requires scale bar and scale value. In addition, the legend and corresponding definition should be provided in the caption of the figure. Moreover, the quality of the figure should be improved.

Modification complete.

5- Fig. 4 and Fig. 5 as well as Fig. 12 require legend and the legend and corresponding definition should be provided in the caption of the figure. In addition the scale bar and scale value of the SEM image is not clear (Fig. 9).

Modification complete.

6- This study suffers from poor quantitative data presentation such as electrochemical impedance spectroscopy measurements and reports the value of charge transfer resistance, solution resistance, constant phase element, and double layer capacitance. Alternatively the author could provide a potentiodynamic polarization test and report the value of corrosion potential, current density, corrosion rate for all samples.

Unfortunately, the potentiodynamic polarization test was not performed in this study. We'll do it in further studies in the future.

7- The author presents a correlation between the composition and corrosion products of the artificial chloride patina during the corrosion process. Further description regarding the measurement of the quantitative measurement of corrosion products and its relation with the composition should be presented. I strongly suggest that the author measure the corrosion rate of all samples by conducting weight loss tests.

The relationship between the quantitative measurement of corrosion products and the components was further explained. Unfortunately, the corrosion rate measurement was not performed in this study. We'll do it in further studies in the future.

8- The author presents the corrosion mechanism of chloride patina in section 4.4. in this regard, For the benefit of the readers the authors could present the schematic corrosion mechanism of the bronze-based material in an aggressive environment.

A schematic diagram of the corrosion mechanism of chloride patina was reflected.

9- A reference about corrosion behavior of Cu-based alloy may be useful for this article: Transactions of Nonferrous Metals Society of China 25 (2015) 1158-1170. In addition, surprisingly small references to the Coatings in the literature despite the large relevant literature there. This should be improved. There are several important papers in recent literature.

Modification complete.

Reviewer 2 Report

This paper conducted artificial copper green corrosion experiments on Cu-Zn-Sn-Pb in order to study the corrosion behavior of copper green, a common chloride in bronzes in the marine environment. It was found that trihydroxy copper chloride was produced early in the corrosion experiments. In addition, the corrosion of copper affected the alloying elements differently, contrary to the corrosion of pure copper. The patina layer is divided into an outer porous powder layer and an inner homogeneous layer. The study is innovative and practical, and provides basic data for the study of sculpture protection and corrosion characteristics such as the color, chemical composition and changes in corrosion products of outdoor bronze sculpture patina surface. Modifications are suggested as follows:
1. It is recommended that the authors add a literature review related to this study in the Introduction section to further reflect the significance of this study.
2. Many of the figures in the article are stitched together from multiple smaller figures, and the author is advised to add small tags to the combined figures.
3. Many of the charts in the article are not neatly arranged, and the author is advised to make adjustments to improve the visualization of the charts in the article.
4. It is recommended that the authors add more rigorous scientific analysis to Figures 8, 9 and 10 to increase the persuasiveness of the article.

5. Writers are advised to check spelling and grammar throughout the text to ensure readability.

Minor editing of English language required

Author Response

Thank you very much for reviewing the article.
I revised and supplemented the parts you reviewed as much as possible.

Review 2 Report (Round 1)

  1. It is recommended that the authors add a literature review related to this study in the Introduction section to further reflect the significance of this study.

Modification complete.

  1. Many of the figures in the article are stitched together from multiple smaller figures, and the author is advised to add small tags to the combined figures.

Modification complete.

  1. Many of the charts in the article are not neatly arranged, and the author is advised to make adjustments to improve the visualization of the charts in the article.

Modification complete.

  1. It is recommended that the authors add more rigorous scientific analysis to Figures 8, 9 and 10 to increase the persuasiveness of the article.

Modification complete. In order to increase the persuasive power of the article, Raman analysis was additionally performed on the cross section of the specimen.

  1. Writers are advised to check spelling and grammar throughout the text to ensure readability.

Modification complete.

Reviewer 3 Report

The author presents a interesting study on the chloride patina of corroding copper alloys. The work is novel and interesting, but requires a revision prior to acceptance.

1.       First sentence has no reference

2.       Second sentence has no reference

3.       Third sentence has no reference

4.       Fourth sentence has ho reference

5.       Fifth sentence has no reference

6.       Page 1 line 39 no reference

7.       Page 1 line 41 no reference

8.       Page 1 line 42 no reference

9.       Page 2 line 45 no reference

10.   Page 2 line 49 no reference

11.   Page 2 line 53 no reference

12.    Why are most references on the topic of this paper and barely any at sentences with claims??

13.   Page 2 line 78 subscript in chemical equation was not performed

14.   Page 3 line 98 please revise

15.   Exact copper composition at start of experiment (blind value) is be needed and also should be stated in methods and materials section

16.   Time for each cycle processing would be helpful

17.   Why is not the thickness change per cycle defined

18.   The author should provide a answer to the corrosion problem, like coating of the statues, like via micro-ark oxidation of surfaces1, or alternaticely sputter coating of the surface2.

Reference

(1)        Kozelskaya, A. I.; Rutkowski, S.; Frueh, J.; Gogolev, A. S.; Chistyakov, S. G.; Gnedenkov, S. V; Sinebryukhov, S. L.; Frueh, A.; Egorkin, V. S.; Choynzonov, E. L.; Buldakov, M.; Kulbakin, D. E.; Bolbasov, E. N.; Gryaznov, A. P.; Verzunova, K. N.; Apostolova, M. D.; Tverdokhlebov, S. I. Surface Modification of Additively Fabricated Titanium-Based Implants by Means of Bioactive Micro-Arc Oxidation Coatings for Bone Replacement. J. Funct. Biomater. 2022, 13 (4), 285. https://doi.org/https://doi.org/10.3390/jfb13040285.

(2)        Akimchenko, I. O.; Rutkowski, S.; Tran, T.-H.; Dubinenko, G. E.; Petrov, V. I.; Kozelskaya, A. I.; Tverdokhlebov, S. I. Polyether Ether Ketone Coated with Ultra-Thin Films of Titanium Oxide and Zirconium Oxide Fabricated by DC Magnetron Sputtering for Biomedical Application. Materials (Basel). 2022, 15 (22), 8029. https://doi.org/10.3390/ma15228029.

Author Response

Thank you very much for reviewing the article.
I revised and supplemented the parts you reviewed as much as possible.

Review 3 Report (Round 1)

  1. First sentence has no reference

Modification complete.

  1. Second sentence has no reference

Modification complete.

  1. Third sentence has no reference

Modification complete.

  1. Fourth sentence has no reference

Modification complete.

  1. Fifth sentence has no reference

Modification complete.

  1. Page 1 line 39 no reference

Modification complete.

  1. Page 1 line 41 no reference

Modification complete.

  1. Page 1 line 42 no reference

Modification complete.

  1. Page 2 line 45 no reference

Modification complete.

  1. Page 2 line 49 no reference

Modification complete.

  1. Page 2 line 53 no reference

Modification complete.

  1. Why are most references on the topic of this paper and barely any at sentences with claims??

It has been supplemented additionally.

  1. Page 2 line 78 subscript in chemical equation was not performed

Modification complete.

  1. Page 3 line 98 please revise

Modification complete.

  1. Exact copper composition at start of experiment (blind value) is be needed and also should be stated in methods and materials section

Modification complete.

  1. Time for each cycle processing would be helpful

It has been supplemented additionally.

  1. Why is not the thickness change per cycle defined

Modification complete. Changes in thickness by cycle were described in sections 3.5 and 4.3.

  1. The author should provide a answer to the corrosion problem, like coating of the statues, like via micro-ark oxidation of surfaces1, or alternaticely sputter coating of the surface2.

We supplemented the coating part additionally.

Round 2

Reviewer 1 Report

Coatings-2595774

Authors have done the part of corrections and suggestions. However, some corrections and explanation must be done previous to publish the paper in the Coatings. Specific questions and comments are listed as follow:

1- This study suffers from poor quantitative data presentation such as electrochemical impedance spectroscopy measurements and reports the value of charge transfer resistance, solution resistance, constant phase element, and double layer capacitance. Alternatively, the author could provide a potentiodynamic polarization test and report the value of corrosion potential, current density, corrosion  

2- Further description regarding the measurement of the quantitative measurement of corrosion products and its relation with the composition should be presented. I strongly suggest that the author measure the corrosion rate of all samples by conducting weight loss tests for all samples.

Author Response

Unfortunately, the specimen that was carried out in this study is not currently available.
Therefore, I don't think the potential mechanical polarizability test and weight loss tests that the reviewer mentioned will be able to be performed.
However, we have completed all the modifications to the other comments you mentioned.
In the future, I will take what you said into my heart and reflect it in other experiments.
I would appreciate it if you could pass this review.